# Receiver Operating Characteristic Curve Analysis of the Somatosensory Organization Test, Berg Balance Scale, and Fall Efficacy Scale–International for Predicting Falls in Discharged Stroke Patients

**DOI:** 10.3390/ijerph19159181

**Published:** 2022-07-27

**Authors:** Iva Fiedorová, Eva Mrázková, Mariana Zádrapová, Hana Tomášková

**Affiliations:** 1Clinic of Rehabilitation and Physical Medicine, University Hospital of Ostrava, 708 52 Ostrava, Czech Republic; mariana.zadrapova@fno.cz; 2Department of Epidemiology and Public Health, Faculty of Medicine, University of Ostrava, 703 00 Ostrava, Czech Republic; eva.mrazkova@centrum.cz (E.M.); hana.tomaskova@osu.cz (H.T.)

**Keywords:** stroke, balance, fall risk assessment

## Abstract

Background: Although fall prevention in patients after stroke is crucial, the clinical validity of fall risk assessment tools is underresearched in this population. The study aim was to determine the cut-off scores and clinical validity of the Sensory Organization Test (SOT), the Berg Balance Scale (BBS), and the Fall Efficacy Scale–International (FES-I) in patients after stroke. Methods: In this prospective cross-sectional study, we analyzed data for patients admitted to a rehabilitation unit after stroke from 2018 through 2021. Participants underwent SOT, BBS, and FES-I pre-discharge, and the fall incidence was recorded for 6 months. We used an area under the receiver operating characteristic curve (AUC) to calculate predictive values. Results: Of 84 included patients (median age 68.5 (interquartile range 67–71) years), 32 (38.1%) suffered a fall. All three tests were significantly predictive of fall risk. Optimal cut-off scores were 60 points for SOT (AUC 0.686), 35 and 42 points for BBS (AUC 0.661 and 0.618, respectively), and 27 and 29 points for FES-I (AUC 0.685 and 0.677, respectively). Conclusions: Optimal cut-off scores for SOT, BBS, and FES-I were determined for patients at risk for falls after a stroke, which all three tools classified with a good discriminatory ability.

## 1. Introduction

Despite significant improvements in treatment, ischemic stroke remains a leading cause of death, disability, impaired body function, and cognitive deficiency [1]. Many neurological consequences of stroke, combined with external and internal environmental factors, can increase the risk of falls and lead to further complications. Falls are a serious problem in all stages after stroke [2], with a one-year post-stroke incidence of 30% to 73% and recurrent falls in 27% of patients [3,4,5]. Falls are associated with injuries of varying severity, decreased physical activity, and functional limitations, leading to the loss of independence and the fear of falling [6]. These additional consequences can entail a longer recovery period and higher treatment costs [7]. Identifying the predictive performance of the most commonly used fall prediction tools may have socioeconomic relevance.

Various predictors, determinants, and circumstances associated with the risk of falling in the acute, subacute, and chronic phases after ischemic stroke have been investigated [8,9,10,11]. Significant risk factors for patients who fall after ischemic stroke include impaired balance, perception, cognitive function, visuospatial hemineglect, a subjective fear of falls, a history of previous falls, depression, the use of walking aids, and impaired self-sufficiency; less common factors include age, incontinence, and gender [4,9,12]. Literature reviews and a meta-analysis have confirmed the multifactorial etiology of falls in stroke survivors, while recommending that risk factors should be identified and addressed by interventions on multiple levels [3,13,14]. There is a need for tools with validated predictive ability in determining fall risk and to assess the effectiveness of the intervention. Previous studies have examined the ability to predict falls after stroke using clinical tests of balance and the fear of falling [4,13,15,16]. However, discriminatory validity and cut-off scores either have not been specified for patients after stroke or need to be updated in a population treated according to current guidelines [12,13].

Balance impairments are a common complication after stroke, leading to falls. Balance disorders occur in approximately 87.5% of stroke patients [14], who also can experience problems with muscle strength, a limited range of motion, impaired coordination, and somatosensory function [14]. Among the sensorimotor consequences, impaired postural control is likely to have the greatest impact on functional independence and gait [15]. Given the variety of neurological deficits and compensatory strategies, an accurate assessment of all aspects of postural control (motor, sensory, and cognitive) is needed [16]. For the subacute and chronic stages, the assessment of postural balance is an important factor in fall risk prediction [17,18]. Multiple standardized tools are available to assess balance in the context of stroke, most commonly clinical evaluation (e.g., the Romberg test, the Dynamic Gait Index, the Functional Reach Test, and the Timed Up and Go), balance scales (the Berg Balance Scale (BBS), the Postural Assessment Scale for Stroke Patients), questionnaires (e.g., the Activities-specific Balance Confidence Scale, the Falls Efficacy Scale), and instrumental posturography (e.g., the EquiTest, NeuroCom, a Force Platform, the Biodex balance system). Several studies have been conducted to validate the most useful assessment tools for predicting falls after discharge. Of these, the BBS has been the most commonly used to predict falls and has been identified in some studies as an effective predictor of falls in patients after stroke [19,20]. However, recent meta-analyses have found considerable heterogeneity among the conducted studies [21,22]. The predictive abilities of BBS for fall risk have been evaluated to date in one meta-analysis, which included studies focusing on risk in patients with Parkinson’s disease or stroke, primarily in the community setting [22]. Predictive values, such as the area under the receiver operating characteristic (ROC) curve, the likelihood ratio (+/−), and positive and negative predictive values have not yet been determined for BBS. Furthermore, with regard to the recency of previous studies, updating the baseline data in relation to predictive abilities is appropriate.

Compared with clinical tests and scales, which carry a bias risk from subjective evaluations by the clinician, instrumental posturography is highly objective [23]. The clinical utility of instrumental posturography as an objective and quantitative measure recommends its use in addition to scales [24,25] and has been confirmed in studies conducted mainly in older people and patients with stroke, Parkinson’s disease, or vestibular disorders [24,25,26]. The choice of appropriate variables and consistent adherence to measurement conditions (standardization) are essential in posturography [27]. The basic test performed within dynamic posturography is the Sensory Organization Test (SOT), which allows for the capture of impaired sensory integration or inadequate sensory reweighting [16]. The SOT has the advantage of allowing for the objective determination of balance parameters such as equilibrium score, somatosensory analysis, and the identification of movement strategies under particular conditions. The few studies that have addressed fall prediction using the SOT have yielded uncertain conclusions. SOT has been studied in a population with vestibular disorders and in people over the age of 65 years [28], but its predictive abilities (equilibrium score) and cut-off scores to identify patients at a high risk of falling after stroke have not been established.

A psychological factor that contributes to an important risk for falls in patients after stroke is self-efficacy related to falls (fear of falling) [29,30]. Approximately 60–70% of patients with chronic stroke report poor self-efficacy related to falls, which is associated with increased anxiety and limitations in mobility balance [30,31]. The Fall Efficacy Scale–International (FES-I) has been developed to derive the level of fear of falling and is a valid tool for discriminating fall risk in patients after stroke [32]. The cut-off value, sensitivity, and specificity of the FES-I for the population requiring rehabilitation in the post-discharge period after stroke have not yet been established. Only one study has identified a cut-off FES-I value in a population of community-dwelling individuals with chronic stroke [32].

The aim of the present study was to determine the performance of the SOT, BBS, and FES-I fall risk-prediction tools, all frequently used in the stroke rehabilitation setting, and establish optimal cut-off scores that could discriminate those at the greatest risk for falls after stroke.

## 2. Materials and Methods

### 2.1. Study Design

This monocentric, prospective, cross-sectional study was conducted in a population of patients admitted for rehabilitation after stroke. The study was approved by the institutional ethics committee (reference number 773/2018). All participants signed informed consent. The study was carried out according to Strengthening the Reporting of Observational Studies in Epidemiology (i.e., STROBE) criteria [33].

### 2.2. Participants and Setting

This study involved patients diagnosed with a primary ischemic stroke admitted to rehabilitation at the Medical Rehabilitation Clinic (CMR) of the University Hospital of Ostrava, Czech Republic, from October 2018 through February 2021. The inclusion criteria were as follows: (1) patients with first ischemic stroke attack < 90 days, (2) age 40–79 years, (3) in stable condition, (4) able to participate, (5) with functional mobility according to Functional Ambulation Categories (FAC) 3–5, and (6) with the ability to stand without support for 5 min. Exclusion criteria were (1) age < 40 or ≥79, (2) decompensated condition, (3) unable to participate, (4) severe phatic disorder, (5) low functional mobility according to FAC 0–2, (6) inability to stand for 6 min, (7) repeated ischemic stroke or death within 6 months after discharge from hospital care, and (8) the presence of an orthopedic or neurological disease that might influence balance.

### 2.3. Clinical Outcome Measures

Demographic and clinical data were obtained from medical documentation (age, sex, time since the stroke, type of stroke, stroke therapy method). The following assessments were applied as baseline measures by trained occupational therapists and physiotherapists 3 days before patient discharge: Montreal Cognitive Test (MOCA), Barthel Index (BI), FAC, SOT, BBS, and FES-I. Patients also were asked one direct yes/no question: “Are you afraid of falling?” The MOCA was used to assess cognitive impairment [34,35] and the BI to assess functional capacity (activities of daily living) [36]. Mobility was assessed according to the FAC scale (0–5) [37].

#### 2.3.1. SOT

SOT was performed using Balance Manager^®^ System (NeuroCom^®^ International, Inc., a Division of Natus, Clackamas, OR, USA; Smart Balance Master^®^, software version 8.6.0), according to the protocol provided with the software, taking into account the age and height of the patient. The ability to stabilize the postural position in upright bipedal standing was tested on a force plate under six different sensorial conditions, as follows: condition 1—fixed surface and visual surround, eyes open; condition 2—fixed surface, eyes closed; condition 3—fixed surface, eyes open, moving, surround; condition 4—moving surface, eyes open, fixed, surround; condition 5—moving surface, eyes closed, fixed, surround; and condition 6—moving surface, eyes open, moving surround.

Each of these conditions was repeated three times, with a duration of 20 s for each record. The resulting values for the center of pressure movement detected by the force platform were compared by the software with the normative values for a healthy population. SOT norms are provided for ages up to 79 years (20–59, 60–69, 70–79) (NeuroCom^®^ International Inc., Clackamas, OR, USA; Smart Balance Master^®^, software version 7.0). These normative values were selected according to the following parameters: height and age, as required by the Smart Balance Master^®^ system software; motor and cognitive status of the normative sample assessed with the Mini-Mental State Examination; the 10-m walking test [38].

The composite score is a value of the arithmetic mean of the percentage of balance obtained in each SOT condition. The resulting point composite (equilibrium score) is an indicator of the stability level: higher values indicate better postural stability, and a drop in values below the 75th percentile indicates instability with a potential risk for falling [39].

#### 2.3.2. BBS

The BBS test consists of 14 items containing principal components for evaluating the ability to maintain balance during standing, sitting, transfers, and rotations that are necessary to perform activities of daily living. The items are evaluated on an ordinal scale of 0 to 4 (0—no performance, 4—highest score), with a maximum of 56 points, and a higher score indicates better stability. The resulting interval scores can be categorized. BBS was originally developed for the prediction of falls in elderly people but is useful for stroke patients, as well [16,22,40,41].

#### 2.3.3. FES-I

The FES-I is a self-report questionnaire and a widely accepted tool to assess perceived confidence in performing a variety of activities of daily living at home and in social settings without falling. The FES-I consists of 16 items scored on an ordinal scale from 1 to 4 (1—not at all concerned, 4—very concerned). Scores range from 16 to 64 points, and higher scores indicate lower self-efficacy [42,43]. In this study, the questions were completed with the patients in the form of structured interviews.

### 2.4. Data Collection Procedure

Balance evaluation (SOT, BBS) and FES-I were administered within 3 days of the planned date of discharge from CMR. Patients and their families were asked to record the date and circumstances of the fall (time, location, association with activities of daily living). A fall was defined as an unexpected and unplanned event in which a person is found on the floor or another horizontal surface, with or without injury [44]. Participants were followed for 6 months until the occurrence of a fall or falls. The dependent variable was any fall after discharge, based on phone-reported data collected every month. For the analysis, we used composite SOT, BBS, and FES-I scores.

### 2.5. Statistical Analysis

Descriptive statistics were used to describe baseline characteristics reporting quantitative data, such as medians and interquartile ranges (IQRs). Participants were classified as ‘non-fallers’ and ‘fallers.’ The differences between these two groups in sociodemographic, clinical, and functional characteristics were calculated using the Mann–Whitney U test for non-normally distributed continuous variables, Pearson’s chi-squared test, and Fisher’s exact test for categorical variables.

The optimal cut-off scores for fall risk classification were calculated using the Youden index for ROC analysis. A ROC curve was plotted for each instrument to verify the test’s ability to identify a fall, using the area under the curve (AUC) as a test scale. To determine the clinical validity of the instruments based on the actual fall, we evaluated sensitivity, specificity, positive and negative predictive values, and positive and negative likelihood ratios.

Statistical analyses were performed using Stata version 13, and the significance level was set at 5%.

## 3. Results

In the period from October 2018 through to February 2021, a total of 1083 patients admitted to CMR were screened, 254 (23.54%) of whom were stroke patients (230 ischemic, 24 hemorrhagic). A total of 128 patients were not included because of not meeting inclusion criteria. The main reasons for exclusion were a low level of functional ambulation based on a FAC score of 0–1 (48; 37.5%), an age > 79 years (39; 30.5%), severe phatic impairment (13; 10.2%), comorbidities that could affect balance (10; 7.8%), recurrent stroke (9; 7%), nonadherence to participation (5; 4.0%), or an age < 40 years (4; 3.1%). This study included 102 patients with ischemic stroke. In total, 18 were lost during the follow-up period (eight individuals were non-adherent, five patients died, and five experienced a stroke recurrence). The remaining 84 patients were eligible for analyses, 42 of them men (50%) and with a median age of 68.5 years (IQR 62.3–73.5). In general, 32 (38.1%; 95% confidence interval (CI) 27.7–49.3) of the 84 patients had a fall or falls within 6 months after discharge. Table 1 presents the sociodemographic, clinical, and functional characteristics of all participants and the results of the comparison between non-fallers and fallers.

There were no statistically significant differences between the two groups in the sex ratio, the time from stroke to measurement, and the affected vascular territory, but they did differ significantly for age and clinical characteristics. The group of patients reporting falls had a significantly higher median age of 72.5 years (IQR 66.0–76.8) vs. 50 years (IQR 55.8–71.0) in the non-falling group; (*p* = 0.011), a cognitive deficit of a median 25 points (IQR 20–27) vs. 27 points (IQR 26–28) in the non-faller group; (*p* = 0.02), a reduced level of independence in activities of daily living with a median BI of 85 points (IQR 75–90) vs. 95 points (IQR 90–100); (*p <* 0.001), and a reduced walking ability with a median FAC of 3 (IQR 3–4) vs. 4 (IQR 3–5) in the non-faller group; (*p* < 0.001). Among the 32 patients who fell, the median FES-I was 36.5 points (IQR 27.0–43.8) vs. 25.5 (IQR 32.3–33.8) in the non-faller group; (*p* = 0.003), and a fear of falling was confirmed in 30 participants (93.8%) by direct questions, confirming a high fear of falling vs. 34 participants (65.4%) in the non-faller group. The results also indicated that people who fell had a significantly altered balance ability, as evidenced by a median SOT of 54 points (IQR 46.0–61.8) vs. 64 points (IQR 53.0–73.4); (*p* < 0.001) and a BBS median of 41 (IQR 31–47) vs. 45 (IQR 40.5–51.0) in the non-faller group; (*p* = 0.003).

Table 2 shows the optimal cut-off scores for SOT, BBS, and FES-I that discriminated between the non-faller and faller groups at discharge. An analysis of the optimal performance cut-off point for the fall risk scale yielded a cut-off of 60 for the SOT, with a sensitivity of 71.9%, a specificity of 65.4%, an AUC of 0.686, and an OR of 4.83. Two cut-off values of 35 and 42 were determined for BBS, with respective sensitivities of 43.8% and 56.3%, specificities of 88.5% and 67.3%, and AUC values of 0.661 and 0.618. For the FES-I, cut-off scores were established at 27 and 29 points, with respective sensitivities of 81.3% and 71.9%, specificities of 55.8% and 63.5%, and AUC values of 0.685 and 0.677. The results of the ROC curve analyses are presented in Figure 1.

To complete the study, baseline predictive values were established for the combination of the instruments studied. Cut-offs with higher AUCs were always used.

(1) In the case of evaluation of the prediction of falls based on all three tools (at least one positive: SOT ≤ 60, BBS ≤ 35, FES-I ≥ 27), the highest sensitivity was found to be 96.9% (95% CI: 83.8–99.9%), but the lowest specificity was found to be 46.2% (95% CI: 32.2–60.5%), AUC = 0.715.

(2) Alternatively, the combination of BBS ≤ 35, FES-I ≥ 27 (at least one positive) was found to have a high sensitivity of 87.5% (95% CI: 71.0–96.5%) and a specificity of 53.8% (95% CI: 39.5–67.8%, AUC = 0.707.

## 4. Discussion

In the present study, we examined the validity of three balance tests to assess the risk of falls in patients after a stroke. The results of the ROC curve analysis showed that cut-off scores of 60 for SOT, 35 and 42 for BBS, and 27 and 29 for FES-I before discharge could be used to predict a decreased fall risk for ambulatory patients after a stroke. According to the ROC analysis AUC, the SOT showed the highest predictive capacity among the tools evaluated.

In our study, 38.2% of patients experienced at least one fall during the first 6 months after discharge from hospital care. These results are consistent with the findings of Samuelson et al., who reported a 23% incidence of falls within the first 3 months and a 40% incidence of falls within one year of stroke [45]. In addition, we find agreement in factors associated with the risk of falling, particularly a low postural control and a fear of falling [4,45]. Other studies report a fall incidence during the first 6 months ranging from 37% to 73%, while confirming a higher risk of falls in later stages after a stroke compared with people who did not experience stroke [6,25,46]. However, unlike our current work, which focused on patients with ambulation ability, these previous investigations did not take into account functional status or mobility, which may be related to fall risk [40]. Studies also are most often focused on community dwelling settings.

Age, cognitive impairment, a functional walking ability, a high fear of falling, and stability impairment as measured by instrumented posturography (SOT) and clinical BBS were significant predictors of falls in post-stroke patients after discharge from rehabilitation. However, 84 patients were from the same cohort of post-stroke patients who are more prone to falls than the healthy reference population [2]. The results may have been influenced by the small sample size and the large variance in values for age and clinical characteristics. It can be hypothesized that, compared to a healthy cohort of patients, the results could be different. Previous studies have documented that falls in poststroke individuals are not related to age, gender, or stroke location [24,46,47]. In our study, the age was a significant predictor of falls. In the faller group was a significantly higher median age of 72.5 vs. a median age of 50 in the non-faller group; the age may have influenced the studied clinical characteristics. A generally older age (over 65 years) is one of the main causes of impaired postural stability, gait, and cognitive function, as well as an increased self-reported fear of falling [8]. The clinical characteristics of both groups confirm postural instability, as demonstrated by a clinical test (BBS) and quantitative posturography (SOT), which are consistent with previous studies that compared the results with a relevant cohort (healthy, same age) [25,48,49]. In the non-faller versus the faller group, postural instability impairment was shown to a lesser extent, which may explain the level of functional categories of gait. Patients who did not suffer a fall had a higher level of ambulation and did not depend on another person for ambulation, reflecting their high level of self-sufficiency as evaluated by the BI. The opposite trend is observed in the faller group, which required the supervision of other people for ambulation and a greater need for the use of a walking aid, resulting in a hypothesized lower level of self-sufficiency. We find consistency with these findings and the relationships regarding impaired stability, gait, and their impact on self-sufficiency in patients after stroke in the studies previously referenced [24,41,50,51,52]. The score of cognitive impairment (MOCA) showed that the faller group had a reduced cognitive function versus the non-faller group. The influence of cognitive functions on stability and self-sufficiency was described in the mentioned studies [46,53,54].

We used dynamic posturography as an objective indicator of balance impairment through the SOT, expressed by the equilibrium score. Only a few studies have focused on using posturography to predict falls in specific at-risk groups. Older adults and patients with Parkinson’s disease and vestibular disorders are the most evaluated populations, and results are heterogeneous and inconsistent, without predictive variables [26,47]. General limitations may include retrospective evaluation, the inconsistent standardization of measurements, and the inappropriate selection of variables [27]. Here, we found that SOT scores were lower in the group with falls than in the group without falls, which is consistent with results of a study in adult patients that primarily assessed stability limits [48]. SOT limits of <38 have been reported in relation to vestibular etiologies; however, ROC values have shown an unsatisfactory sensitivity of 53% and a specificity of 87% [26]. A prospective study to determine a cut-off score that discriminates between patients who do and do not experience falls has not been performed, precluding the comparison of our findings with previous results. The main advantage of posturography is the possibility to objectivize postural stability even in patients with cognitive deficits, who are difficult to administer a questionnaire to (e.g., FES-I). From a practical point of view, posturography has economic disadvantages because it requires instrumentation. Nevertheless, given the current trend in technology development, the development of alternative devices that could be supported to perform on a similar principle. For clinical evaluation, we assessed the BBS, which is popular for its low demand in terms of equipment and administration work. However, the results of previous studies show heterogeneity and some important limitations with this tool. Lima et al. conducted a systematic review of BBS for making fall predictions in elderly adults. They were unable to perform a meta-analysis because of heterogeneity among studies, and based on insufficient evidence to support the use of BBS for falls prediction, they concluded that the instrument should not be used alone in older patients [21]. In contrast, in a meta-analysis of 33 articles, Kudlac et al. evaluated the predictive ability of falls BBS and found that it had excellent reliability and validity. They concluded that the BBS is reliable and valid for assessing balance and functional mobility in the post-stroke population; however, this tool should not be considered a strong predictor of fall risk in this patient group [49]. Park et al. conducted a meta-analysis of data from 21 studies describing the predictive validity of BBS for fall risk and reported a high statistical heterogeneity among them. Regarding the overall predictive validity of the BBS, they found a pooled sensitivity of 0.72, a specificity of 0.73, and an AUC of 0.84. In subgroup analyses, an age < 65 years, neuromuscular disease, two-plus falls, or a cut-off score of 45–49 were associated with better sensitivity with statistically less heterogeneity [30]. In our study, the BBS scores in the group without falls were very low compared with the group that experienced falls. Although statistical software has established two cut-off values based on the calculation of the Youden index, both provide low sensitivity but higher specificity, which is of little clinical use. The AUC value was the lowest compared to the other tools studied. However, it can be judged sufficiently discriminatory in the 0.6–0.7 interval [55,56], where otherwise all the tools studied fall.

We found a statistically significant fear of falls among those who experienced falls versus those who did not, which we confirmed by asking a direct yes/no question and using the FES-I questionnaire. The fear of falling was confirmed by direct questioning in the non-falling group by 34 patients (65.4%) compared to the faller group, where 30 individuals (93.8%) confirmed the fear, which was also evident from the FES-I, where patients in the faller group reported a high fear of falling. The FES-I questionnaire showed high sensitivity values and low specificity. The AUC values for the established cut-offs were similar to those of the SOT and can be assessed as indicating borderline low accuracy. In a study of patients with chronic stroke, Faria-Fortini et al. found a FES-I cut-off of 28 points for distinguishing those with falls from those without (AUC = 0.71; sensitivity = 71%; specificity = 57%; positive predictive value = 51%; negative predictive value = 74%) [32]. The main limitation of the FESI questionnaire is its use in people with a cognitive impairment; there is a risk of bias.

In this study, we determined the baseline predictive values for the combination of two or more tests using the cut-off point with the highest AUC. Prediction of the fall based on all tools (SOT, BBS, FES-I—at least one positive) increased sensitivity, but the results were not statistically significant compared to the results of SOT.

Our study has some limitations. First, the main limitation is the small sample size, which causes a large variance of values. Second, the study analyzed only 84 patients from the same identical cohort of stroke patients who were discharged from rehabilitation. Third, we did not include nonambulatory people and people with mild symptoms at discharge from acute unit hospital care. These features were determined based on inclusion criteria in the posturographic examination and on the assumption that nonambulant patients would have a low probability of falling. Third, we did not assess repeated falls for this study.

A strength of our study is the confirmation of the idea that impaired stability and a fear of falling precede falls. The results provide optimal cut-off scores for SOT, BBS, and FES-I in post-stroke patients, which may be useful in preventing falls, evaluating intervention effectiveness, and providing care through telemedicine.

## 5. Conclusions

SOT, BBS, and FES-I demonstrated a good discriminatory ability to classify post-stroke patients who were fallers from non-fallers. Of the tools evaluated, SOT showed the most promise in predicting falls. The results of our study support the recommendation to use multiple tools that assess static balance and dynamic balance, as well as functional tests and the consideration of fear of falls when predicting fall risk in patients after stroke. The accurate assessment of fall risk and the ability to maintain balance is important to understand the extent and type of balance deficit and provide effective customized preventive interventions. The results may be useful in designing fall prevention interventions for patients after stroke.

## Figures and Tables

**Figure 1 ijerph-19-09181-f001:**
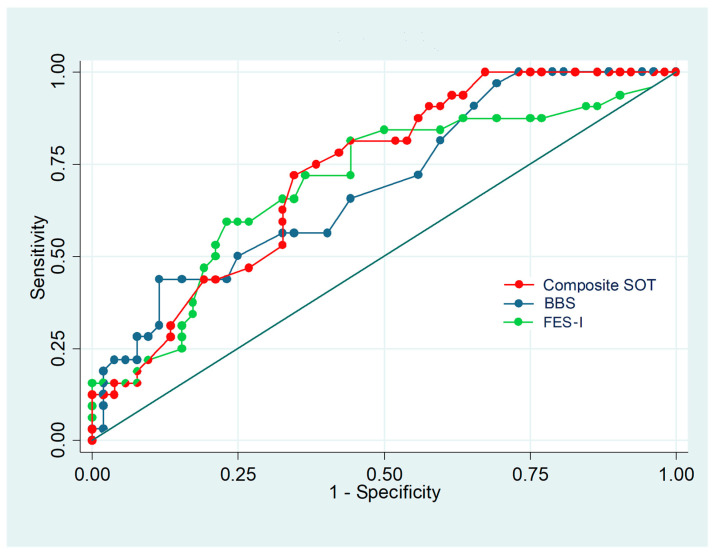
Receiver operating characteristic curve analysis of the equilibrium score of the Sensory Organization Test (composite SOT), the Berg Balance Scale (BBS), and Falls Efficacy Scale–International (FES-I).

**Table 1 ijerph-19-09181-t001:** Demographic and clinical characteristics of the study participants (N = 84).

Characteristics	All Participants N = 84	No. Falls within 6 Months (Non-Fallers) N = 52	Falls within 6 Months (Fallers) N = 32	*p*
Age (years), median (IQR) *	68.5 (62.3–73.5)	50.0 (55.8–71)	72.5 (66–76.8)	0.011
Sex, female (%) **	42 (50)	22 (42)	20 (62)	0.072
Time from the onset of stroke to measurement (day), median (IQR) **	22 (11–33)	21 (17–30)	23 (19–35)	0.233
Affected vascular territory				
ACM, *n* (%) **	35 (41.7)	22 (42.3)	13 (40.1)	0.879
ACA, *n* (%) **	5 (6.0)	2 (3.9)	3 (9.4)	0.298
VB, *n* (%) **	27 (32.1)	18 (34.6)	9 (28.1)	0.536
Cerebellum, *n* (%) ***	11 (13)	9 (17.3)	2 (6.23)	0.193
Arteria carotis int, *n* (%) ***	8 (9.5)	3 (5.8)	5 (15.6)	0.249
Clinical characteristics				
MOCA (score), median (IQR) *	27 (23–28)	27 (26–28)	25 (20–27)	0.020
BI (score), median (IQR) *	90 (85–95)	95 (90–100)	85 (75–90)	<0.001
FAC, median (IQR) *	4 (3–4)	4 (3–5)	3 (3–4)	<0.001
Fear of falling, *n* (%) **	64 (76.2)	34 (65.4)	30 (93.8)	0.003
SOT (score), median (IQR) *	61 (48.0–70.8)	64 (53.0–73.4)	54 (46.0–61.8)	<0.001
BBS (score), median (IQR) *	45 (37.0–48.8)	46 (40.5–51.0)	41 (31.0–47.0)	0.003
FES-I (score), median (IQR) *	28.5 (23.0–38.8)	25.5 (21.3–33.8)	36.5 (27.0–43.8)	0.003

Legend: VB, vertebrobasilar basin; ACM, arteria cerebri media; ACA, arteria cerebri anterior; MOCA, Montreal Cognitive Assessment; BI, Barthel Index; FAC, Functional Ambulation Categories; SOT, Somatosensory Organization Test; BBS, Berg Balance Scale; FES-I, Fall Efficacy Scale–International; * Mann–Whitney U test; ** Pearson chi-squared test; *** Fisher’s exact test.

**Table 2 ijerph-19-09181-t002:** The optimal cut-off scores for SOT, BBS, and FES-I using the Youden method.

Cut-Off Score	SOT ≤60	BBS ≤35	BBS ≤42	FES-I ≥27	FES-I ≥29
	Value	Value	Value	Value	Value
	95% CI	95% CI	95% CI	95% CI	95% CI
Prevalence (%)	32/84 38.1	32/84 38.1	32/84 38.1	32/84 38.1	32/84 38.1
	(27.7–49.3)	(27.7–49.3)	(27.7–49.3)	(27.7–49.3)	(27.7–49.3)
Sensitivity (%)	23/32 71.9	14/32 43.8	18/32 56.3	26/32 81.3	23/32 71.9
	(53.3–86.3)	(26.4–62.3)	(37.7–73.6)	(63.6–92.8)	(53.3–86.3)
Specificity (%)	34/52 65.4	48/52 88.5	35/52 67.3	29/52 55.8	33/52 63.5
	(50.9–78.0)	(76.6–95.6)	(52.9–79.7)	(41.3–69.5)	(49.0–76.4)
PPV (%)	56.1	70.0	51.4	53.1	54.8
	(39.7–71.5)	(45.7–88.1)	(34.0–68.6)	(38.3–67.5)	(38.7–70.2)
NPV (%)	79.1	71.9	71.4	82.9	78.6
	(64.0–90.0)	(59.2–82.4)	(56.7–83.4)	(66.4–93.4)	(63.2–89.7)
LR (+)	2.1	3.8	1.7	1.8	2.0
	(1.34–3.2)	(1.6–8.9)	(1.1–2.8)	(1.3–2.6)	(1.3–3.0)
LR (−)	0.43	0.64	0.65	0.34	0.44
	(0.24–0.77)	(0.46–0.88)	(0.42–1.01)	(0.16–0.72)	(0.25–0.8)
AUC	0.686	0.661	0.618	0.685	0.677
	(0.58–0.79)	(0.56–0.76)	(0.51–0.73)	(0.59–0.78)	(0.57–0.58)

Legend: SOT, Sensory Organization Test; BBS, Berg Balance Scale; FES-I, Falls Efficacy Scale–International; CI, confidence interval; PPV, positive predictive value; NPV, negative predictive value; LR, likelihood ratio; AUC, area under receiver operating characteristic curve.

## Data Availability

Data are available on request to the corresponding author.

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
