# Peer review of "Receiver Operating Characteristic Curve Analysis of the Somatosensory Organization Test, Berg Balance Scale, and Fall Efficacy Scale–International for Predicting Falls in Discharged Stroke Patients"

_ijerph, 2022, doi:10.3390/ijerph19159181_

Round 1
Reviewer 1 Report
The article characterized the cut-off score for three risk assessment tools for post-stroke falling.
The major concern is that there are only 84 patients. Given such small sample size, the cut-off score estimation should have a very large variance. However, such variance is not adequately discussed in the article.
Meanwhile, the calculation of p-value is most meaningful when the samples are generated from a random process. However the 84 patients came from the same cohort and thus expected to be correlated. Estimation of a correlated sample tends to be associated with high variance. Today a sample of 84 people has a small p value between fallers and non-fallers, tomorrow another sample of 84 people may have a large p value. The correlation within the fallers and non-fallers are not thoroughly discussed.
Author Response
The corrections are provided in the Annex.

Reviewer 2 Report
The manuscript by Fiedorova et al evaluates the efficacy of sensory organization test (SOT), Berg balance scale (BBS), and fall efficacy scale international (PESI) tools to predict the occurrence of falls in post-stroke patients. The patients were followed for three years post-stroke and fall predictive values were calculated using area under the ROC curve (AUC) method. Authors find that all of these tools are effective in predicting post-stroke patients who were fallers from non-fallers. The highest predictive efficacy was found for SOT.
The manuscript is well written with clear background information. Data are well analyzed and conclusions supported by the data. The limitations of the study have also been explicitly mentioned.
Author Response

(The authors gave the same response as above.)

Reviewer 3 Report
The authors tested the validity of various clinical scales (Berg Balance Scale, Fall Efficacy Scale) evaluated by physicians and a quantitative test (Somatosensory Organization Test) in predicting falls on discharged stroke patients. The clinical evaluations were administered before discharging the subjects, and participants falls were recorded for the next 6 months.
In my opinion this is a well written paper with grounded methodology. I have no particular comments on the selection of participants, inclusion and exclusion criteria, sample size and test assessments.
My main concern, that I think should be addressed in the discussion, is related to the usefulness and the interaction of the various tests. In my understanding, while every clinical test, within this population (MoCA and Barthel included), is more or less useful to predict the possibility of falling; even the most accurate, SOT, has a sensitivity and specificity of less than 75%. I think it is important to at least discuss the statistical interaction that two or more test have on improving the prediction accuracy of preventing falls. My rationale is also regarding SOT, as it is a really an informative technology but requires specialized equipment that could not be economically viable for a simple screening of the population.
Minor remarks:
- Please check the tables for minor typos and missing information (e.g. years and BI acronym not explained in the legend in table 1)
- In Methods page 4 on line 154, it is stated that the measurements of sway are compared with normative values for a healthy population. Is it possible to cite the source? Also, these normative values were selected accordingly with some parameters (age, height or sex)?
Author Response

(The authors gave the same response as above.)
